# How Distinguishable Are Vocoder Models? Analyzing Vocoder Fingerprints for Fake Audio

## Abstract

In recent years, vocoders powered by deep neural networks (DNNs) have found much success in the task of generating raw waveforms from acoustic features, as the audio generated becomes increasingly realistic. This development however raises a few challenges, especially in the field of forensics, where the attribution of audio to real or generated sources is vital. To our knowledge, our investigation constitutes the first efforts to answer this question on the existence of vocoder fingerprints and to analyze them. In this paper, we present our discoveries in identifying the sources of generated audio waveforms. Our experiments conducted on the multi-speaker LibriTTS dataset show that (1) vocoder models do leave model-specific fingerprints on the audio they generate, and (2) minor differences in vocoder training can result in sufficiently different fingerprints in generated audio as to allow for distinguishing between the two. We believe that these differences are strong evidence that there exist vocoder-specific fingerprints that can be exploited for source identification purposes.

## 1 Introduction

Over the past decades, advancements in text-to-speech (TTS) and voice conversion (VC) technologies permit the artificial generation of speech audio. As the last step in TTS or VC pipelines, generating raw waveforms from acoustic features, vocoders are important components in these tasks. With the development of deep learning, deep neural networks (DNNs) have seen increasing usage in both TTS and VC. Recent advancements in these areas (Wang et al., 2017; Shen et al., 2018; Kalchbrenner et al., 2018; Li et al., 2019; Ren et al., 2022) have led to the capacity of generating increasingly realistic and natural audio. While such technological progress is certainly welcomed, it does also admittedly come with challenges, especially when one considers the potential misuse and abuse of TTS and VC technologies, either inadvertently or maliciously. The risks and challenges associated with these technologies mostly arise in the following areas:

**Audio forensics.** As TTS technologies generate increasingly realistic audio, they also become increasingly prone to being misused, by their very nature, to mislead or even defraud victims who cannot judge the authenticity of audio. The possibility of misuse and abuse evidently gives rise to concerns about security, and public order. Therefore, it is consistent with the public interest to find methods to combat such risk of malicious usage by coming up with effective forensic strategies. While modern audio forensic techniques used in challenges such as ASVspoof (Wu et al., 2015; Wang et al., 2020; Yamagishi et al., 2021) and ADD (Yi et al., 2022) have achieved effective results in separating fake audios from real ones, they focus more on performing binary classification between real and fake sets. There is nonetheless also an interest in surpassing the constraints of binary real/fake classification and actually pinpoint the source responsible for generating any fake audio.

**Intellectual property protection.** Mainstream TTS and VC systems such as Shen et al. (2018), Arik et al. (2018), and Ren et al. (2022) are built with much effort, as they require careful design and fine-tuning of models and algorithms, as well as extensive training with more data than can reasonably be collected by any one person. As such, it is in the interest of researchers and commercial software developers alike to ensure that they are properly recognized to hold the ownership and copyright of their intellectual properties, the enforcement of which would not be feasible without the

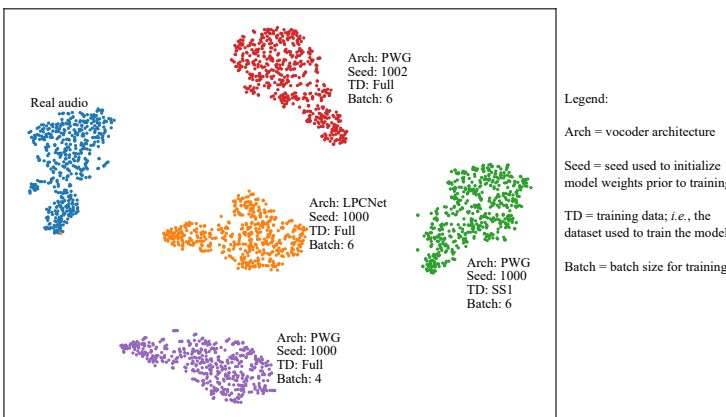

Figure 1: A t-SNE projection of vocoder fingerprints belonging to models of different architectures, initialization seeds, and hyperparameters, obtained in our experiments. ("PWG" refers to the Parallel WaveGAN architecture (Yamamoto et al., 2020); "Full" refers to the clean subset of LibriTTS (see section 4.1); SS1 consists of 50,000 randomly selected samples from the "Full" set)

ability to detect unauthorized usage of a system. While there have been efforts in watermarking a given DNN model (Zhang et al., 2018), these efforts require crafted inputs on models that have been specifically tweaked for watermarking during training, instead of relying on properties intrinsic to the vocoder model in performing waveform generation.

Owing to the reasons above, it is important to be able to identify the source of fake audio. Given the advancement of TTS and VC technologies, and judging by the results obtained in work by other scholars, where generated audios are often judged to be quite natural and realistic, we consider it infeasible to manually perform such scrutiny and source identification. Consequently, this calls for a way to automatically extract relevant information from audio samples and classify fake audio based on its source of generation, with minimal human intervention.

While the idea of source-dependent fingerprints have been explored on GAN-generated images (Yu et al., 2019; Marra et al., 2019), those efforts rely on image properties and are not directly applicable to waveforms generated by vocoders. While there has been some preliminary investigation on the concept of vocoder fingerprints (Yan et al., 2022), the effort of the study rests on an architecture-specific level. We seek to find the existence of fingerprints that allow for distinguishing models sharing the same architecture, separating audios generated by different vocoder models, and identifying their respective sources of generation.

In this paper, we put forth and seek to explain the following questions:

1. Do vocoder models leave model-specific fingerprints on the audio they generate?
2. What is the condition for their existence? In other words, how similar can two vocoder model instances be before it becomes infeasible to distinguish between the two?
3. How do we present these fingerprints for the purposes of fake audio source identification?

To answer these questions, we perform experiments to evaluate our hypothesis that vocoder models used to generate audio leave a fingerprint on the audio they generate that is specific to the model, as well as to explore how these fingerprints may be visualized and exploited for the aforementioned purposes of audio forensics and intellectual property protection. Our experiments confirm the existence of these fingerprints and show that vocoder fingerprints depend on architecture, training data, initialization seeds, and training hyperparameters, as shown in Figure 1.

## 2 RELATED WORK

As our work deals with finding vocoder-specific features to aid in fake audio source identification, a task that falls in the purview of forensics, it behooves us to reiterate some of the work done by other scholars in the areas of vocoders and audio forensics.

Vocoders refer to signal processing devices that synthesize speech audio waveform from some more compressed form of data, usually acoustic features, although certain vocoders take linguistic features as inputs (Tan et al., 2021). Apart from some modern end-to-end systems that directly generate waveform from text, vocoders form the last step in TTS and VC. Notable traditional (*i.e.*, non-neural) vocoder implementations include: Griffin-Lim (Griffin & Lim, 1984), STRAIGHT (Kawahara, 2006), and WORLD (Morise et al., 2016).

With the development of deep learning and neural networks came a variety of neural vocoders, of which the notable ones include: WaveNet (van den Oord et al., 2016; 2018), WaveRNN (Kalchbrenner et al., 2018), LPCNet (Valin & Skoglund, 2019a;b). Given the length of waveforms, autoregressive waveform generation takes much inference time; therefore modern neural vocoders have taken a non-autoregressive approach with generative models, among which are FloWaveNet (Kim et al., 2019), Parallel WaveGAN (Yamamoto et al., 2020), HiFiGAN (Kong et al., 2020), Multi-band Mel-GAN (Yang et al., 2020), Style-MelGAN (Mustafa et al., 2021). These non-autoregressive models are able to achieve faster-than-real-time inference speeds.

One of the primary aspects of audio forensics, known as "integrity verification", deals with the determination of authenticity of a given audio clip, *i.e.*, if the given audio represents real speech and if it has been tempered with (Maher, 2009; Zakariah et al., 2018). Various techniques have been used to analyze details and features contained in the audio, such as the Electronic Network Frequency (Nicolalde-Rodriguez et al., 2010), background noise (Malik, 2013), recording device (Qamhan et al., 2021), or energy (Luo et al., 2018).

In recent years, there has been significant progress in integrity verification of fake audio. Many challenges have accelerated the progress and research in this area, including the ASVspoof challenges (Wu et al., 2015; Kinnunen et al., 2017; Wang et al., 2020; Yamagishi et al., 2021). In addition to these, ADD 2022 (Yi et al., 2022) initiated the first audio deep synthesis detection challenge for more challenging realistic scenarios. In summary, the research on fake audio detection can be broadly divided into two parts: front-end features and back-end classifiers. In the first category, many features (Todisco et al., 2017; Davis & Mermelstein, 1980; Sahidullah et al., 2015) have been widely used for detection. As for the back-end classifier, many effective classifier designs (He et al., 2016; Bagherinezhad et al., 2017; Hu et al., 2018; Tak et al., 2021)have contributed to the detection research.

## 3 METHODOLOGY

### 3.1 VOCODER FINGERPRINT

To answer the questions we put forth, we study the impact of each of the following aspects of vocoder models:

1. **Vocoder architecture:** We procure 7 models, each with a different architecture, ranging from traditional signal processing vocoders to more modern GAN-based vocoders, and study the separability of the fingerprints left by these models on the audio they generate. The architectures are chosen to represent both traditional and neural vocoders. Specifically, WORLD is a traditional signal processing vocoder that achieves promising results; LPCNet utilizes linear prediction, and achieves considerable performance despite the limitations of autoregressive vocoders; the GAN-based vocoders are chosen for their performance reported in their respective papers, achieving faster-than-real-time inference speeds given their nonautoregressive nature.

2. **Training data:** In the training process, each training sample may differently influence the direction to optimize the objective function, hence result in slightly different weights when the model converges. To study the impact of training data, we train 4 models of the same architecture with the same initialization seed, but each model is trained with a slightly different training set that differ from each other in different magnitudes (see Table 4), and analyze the fingerprints left by the models.

3. **Initialization seeds:** Initialization seeds determine the initial weight values before training begins, and thus can potentially subtly influence the resulting weights as the model converges. To examine the impact of initialization seeds, we train 4 models of the same

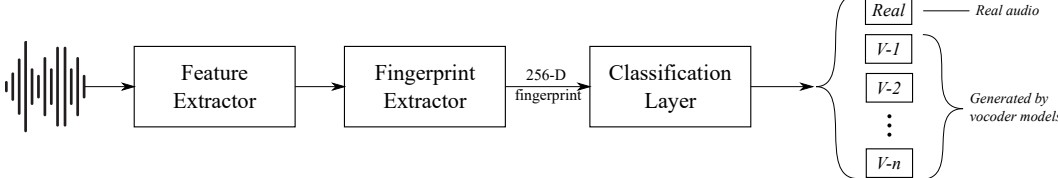

Figure 2: The overall fingerprint extraction and classification pipeline.

architecture, with the exact same training set, but each model is initialized with a different seed. We then analyze the fingerprints left by the models on the generated audio.

4. **Training hyperparameters:** We train 5 models of the same architecture, with the same training set and initialization seed, but adjust the dropout rate and batch size used in the training of each model. We then analyze the fingerprints left by the models on the audio. The choice of dropout rate and batch size are justified because dropout rate, by "freezing" certain weights, can make the vocoder models less "reactive" to incoming training samples, while batch sizes influence the order in which the samples contribute to the tweaking of model parameters.

5. **Architecture and parameter differences, combined:** We train 3 models of 4 GAN vocoder architectures each, varying in seed and training hyperparameters. We then copy-synthesize audio using the vocoders such that the testing audio is generated by vocoders whose parameters are unknown to the classification network even though the architecture is known. We examine the performance of the classification network in this scenario.

## 3.2 VOCODER MODEL

Investigating the existence of vocoder fingerprint requires focus on vocoder-specific information and features, and as such, all other variables need to be fixed, with the vocoder model used to generate audio waveforms being the only independent variable in our experiments. To accomplish this, we generate fake audio by performing copy synthesis on real audio samples from the LibriTTS dataset. This is done by extracting acoustic features from these real audio samples and passing the extracted features through trained vocoder models to generate new waveforms, which will then be used for fingerprint extraction classification. The purpose behind this design is to eliminate any risk of potential interference that could be induced by acoustic models. Furthermore, since mainstream TTS and VC pipelines like Tacotron 1/2 (Wang et al., 2017; Shen et al., 2018) and FastSpeech 2 (Ren et al., 2022) still rely on vocoder models as the final step to generate audio waveforms, we think that vocoder differences are the most likely to cause fingerprint differences.

In addition, samples in the real audio corpus exhibit some variation in terms of speech content and speaker characteristics, the effect of which should not be relied on for the purpose of source identification; evidently, the text or even phonemic content of sentences being spoken as little to do with vocoders. To account for and preclude the impact of this naturally-occurring variance, the generated audio waveforms from each model mirror and correspond to each other as well as real audio waveforms. More specifically, for every real audio sample $x$, we generate fake audio samples $\{\hat{x}_m \mid m = 1, \ldots, M\}$ using each of the $M$ models studied in the experiment, so that groups of real-and-fake audio waveforms from the same acoustic features maximally balance out the effect of the naturally occurring variations intrinsic in the real audio corpus.

## 3.3 FINGERPRINT EXTRACTION PIPELINE

As the motivation behind fingerprint extraction and analysis is essentially to facilitate generation source identification of fake audio, the fingerprints themselves should be such that audio samples from a single source — either real recordings or generated by a single vocoder model — form distinct, preferably linearly-separable, clusters in the feature space that can then be easily distinguished even with a single fully-connected layer (FC layer). As such, we train our fingerprint extraction network essentially as part of a classification model.

Table 1: Datasets used in our experiments.

(a) Dataset used in our experiments to train vocoder models: all audio comes from LibriTTS. This table is excerpted from Zen et al. (2019).

| Subset | Hours | Speakers | | | Utterances |
| | | Female | Male | Total | |
|---|---|---|---|---|---|
| *train-clean-100* | 53.78 | 123 | 124 | 247 | 33,236 |
| *train-clean-360* | 191.29 | 430 | 474 | 904 | 116,500 |
| *dev-clean* | 8.97 | 20 | 20 | 40 | 5,736 |
| *test-clean* | 8.56 | 19 | 20 | 39 | 4,837 |
| Total | 262.60 | 592 | 638 | 1,230 | 160,369 |

(b) Real audios used for copy synthesis to generate training data for fingerprint extraction–classification.

| Subset | Speakers | Utterances |
|---|---|---|
| *train* | 1129 | 50,000 |
| *dev* | 40 | 5,000 |
| *test* | 39 | 4,500 |
| Total | 1208 | 59,500 |

As shown in Figure 2, Each audio sample is first passed through a feature extractor to extract its LFCC features, and the features are then fed into the fingerprint extractor, which is a Res2Net network. The network produces a vector for every input, and the vector is then passed into a FC classification layer to be classified into a finite number of labels, representing the source of the audio (either real or the model that generated it).

## 4 EXPERIMENTS AND RESULTS

In this section, we present our experiments as well as the results and observations obtained. We perform four experiments, each of which focuses on one independent variable, while the rest are controlled and kept the same across models. The vocoder models are trained following the setup in their original paper as closely as possible.

For each audio waveform sample, we first extract its linear frequency cepstral coefficient (LFCC) representation, with 20 LFCC per frame, using a Hamming window with the window length of $M = 480$ and hop length of $H = 240$. The LFCC matrix is then truncated or 0-filled to contain 500 frames. The LFCC representation of the audio sample is then passed to a Res2Net network (Gao et al., 2021) and projected onto a feature space with a dimensionality of 256, then passed through a single FC layer classifier to decide between the finite number of sources. The entire network (extraction + classification) is trained using cross-entropy loss.

### 4.1 DATASET

In our experiments, we use the LibriTTS dataset (Zen et al., 2019) to train the neural vocoder models as well as to generate the audio corpus for vocoder fingerprint extraction network through copy synthesis. We employ only the clean speech subset of the dataset, as shown in Table 1a. With the exception of the second experiment described below in section 4.4, the clean training set of LibriTTS is used in its entirety to train the neural vocoder models.

To train our extraction–classification network in each experiment, we randomly sample 50,000 samples from the aforementioned LibriTTS training set, 5,000 samples from the validation set, and 4,500 samples from the testing set, as shown in Table 1b, to serve as the real audio portion of the classifier training set the basis to perform copy synthesis. The fake audio part thus mirrors the real audio portion, with each class essentially mirroring the real audio set.

### 4.2 EVALUATION METRIC

As discussed in section 3.3, the performance of fingerprint extraction and analysis should be directly related to the ability to classify the audio samples and identify their source of generation, as such, we evaluate our fingerprint extraction model in a way similar to classifier evaluations: by using precision, recall, and F1 scores.

### 4.3 IMPACT OF VOCODER ARCHITECTURES

In our first experiment, we test for the existence of architecture-specific fingerprints in the generated waveforms. We acquire models of the following 6 architectures:

Table 2: Results (%) on the impact of vocoder architectures (4.3)

|  | Precision | Recall | F1 score |
|---|---|---|---|
| REAL | 98.30 | 97.49 | 97.89 |
| PWG | 99.42 | 99.87 | 99.65 |
| HIFIGAN | 97.73 | 98.36 | 98.04 |
| MB_MELGAN | 99.62 | 99.56 | 99.59 |
| S_MELGAN | 99.73 | 99.82 | 99.78 |
| WORLD | 99.80 | 99.58 | 99.69 |
| LPCNET | 100.00 | 99.93 | 99.97 |
| Average | 99.23 | 99.23 | 99.23 |

Table 3: Results (%) on the impact of training data (4.4)

|  | Precision | Recall | F1 score |
|---|---|---|---|
| REAL | 99.78 | 99.58 | 99.68 |
| SS0 | 93.14 | 92.56 | 92.84 |
| SS1 | 92.79 | 95.58 | 94.17 |
| SS2 | 97.76 | 94.09 | 95.89 |
| SS3 | 93.94 | 95.42 | 94.68 |
| Average | 95.48 | 95.44 | 95.45 |

- **WORLD** (Morise et al., 2016): estimates the fundamental frequency (F0), spectral envelope, and aperiodic parameters, as well as generates speech similar to some input speech based on the estimated features.

- **LPCNet** (Valin & Skoglund, 2019a): a neural vocoder combining linear prediction and recurrent neural networks (RNNs) and generates audio waveforms.

- **Parallel WaveGAN** (PWG; Yamamoto et al., 2020): a neural vocoder that generates audio from Mel spectrum using a generational adversarial network (GAN).

- **HiFiGAN** (Kong et al., 2020): a GAN-based neural vocoder that employs two discriminators to perform multi-scale and multi-period discriminators.

- **Multi-band MelGAN** (Yang et al., 2020): a GAN-based neural vocoder with a generator that produces sub-band signals from which the full-band signals are constructed.

- **Style-MelGAN** (Mustafa et al., 2021): a GAN-based neural vocoder that allows for the synthesis of high-fidelity speech with relatively low computational complexity.

We follow the setup instructions in ParallelWaveGAN and attempt to unify the spectrum input to the different GAN vocoder models, as to remove the influence of the input features in the vocoders. The GAN models all take 80-band mel-spectra as inputs, with a frame length of 50 ms and a hop length of 12.5 ms; as a result, these features are pre-extracted and remain the same for all GAN models. The LPCNet vocoderis trained directly on the raw waveforms, since it has built-in feature extractors.

We then tag the models as WORLD, LPCNET, PWG, HIFIGAN, MB_MELGAN and S_MELGAN, respectively. Other than WORLD, which does not require training, each model is trained with the entirety of the LibriTTS clean speech subset described in section 4.1. The models are each trained on a single Tesla K80 GPU, with their default hyperparameters according to their respective papers, except the training batch size of Style-MelGAN was lowered to 16 for the sake of memory usage.

We then train the extraction–classification network described in section 3.3 to distinguish between audio waveforms from 7 classes: {REAL, WORLD, LPCNET, PWG, HIFIGAN, MB_MELGAN, S_MELGAN}. To this end, we sample 50,000 waveform files from each source for training and 5,000 files for validation.

In this experiment, we obtained results reported in Table 2 and Figure 4, showing that models with different architectures produce clearly distinguishable audios. The only relative confusion in this experiment is between the original (real) audio and the audio generated with HiFiGAN, as all other misclassifications constitute less than 1% of the evaluation set. Furthermore, if we were to evaluate the separability between generated audios from different models, no misclassification amounts to more than 0.5% of all audio samples of its real class. As different vocoder architectures take different approaches to construct audio waveforms from spectrograms or other feature representations, this separability is to be expected.

## 4.4 IMPACT OF TRAINING DATA

In our second experiment, we explore the relationship between the training dataset used to train the vocoder model and its fingerprint on generated waveforms. We fix the models to a single GAN architecture: Parallel WaveGAN (Yamamoto et al., 2020), and use slightly different training datasets

Table 4: Difference (in utterances) of training data used in the second experiment

|      | SS0    | SS1    | SS2    | SS3    |
|------|--------|--------|--------|--------|
| SS0  | -      | 1      | 100    | 10,000 |
| SS1  | 1      | -      | 101    | 10,001 |
| SS2  | 100    | 101    | -      | 10,100 |
| SS3  | 10,000 | 10,001 | 10,100 | -      |

Table 5: Vocoder models with different hyper-parameters

|          | Dropout | Batch size | # Steps trained |
|----------|---------|------------|-----------------|
| BASE     | 0       | 6          | 500,000         |
| 1DROPOUT | 0.1     | 6          | 500,000         |
| 2DROPOUT | 0.2     | 6          | 500,000         |
| 4BATCH   | 0       | 4          | 750,000         |
| 8BATCH   | 0       | 8          | 375,000         |

Table 6: Results (%) of on the impact of initialization seeds (4.5)

|           | Precision | Recall | F1 score |
|-----------|-----------|--------|----------|
| REAL      | 99.82     | 99.84  | 99.83    |
| SEED-1000 | 94.51     | 94.91  | 94.71    |
| SEED-1001 | 94.49     | 94.42  | 94.45    |
| SEED-1002 | 94.37     | 97.16  | 95.74    |
| SEED-1003 | 95.52     | 92.33  | 93.90    |
| Average   | 95.74     | 95.73  | 95.73    |

Table 7: Results (%) of on the impact of hyperparameters (4.6)

|          | Precision | Recall | F1 score |
|----------|-----------|--------|----------|
| REAL     | 99.76     | 99.84  | 99.80    |
| BASE     | 97.38     | 94.93  | 96.14    |
| 1DROPOUT | 97.70     | 97.07  | 97.38    |
| 2DROPOUT | 99.12     | 98.09  | 98.60    |
| 4BATCH   | 95.66     | 97.96  | 96.79    |
| 8BATCH   | 95.08     | 96.71  | 95.89    |
| Average  | 97.45     | 97.43  | 97.43    |

to train them. This variance is controlled by sampling 50,000 waveform files from the LibriTTS clean speech set for training and 5,000 waveforms for testing. We tag this base set as SUBSET-0. Three more training subsets of the same sizes are then constructed, each differing from the base set by replacing 1, 100, and 10000 samples, respectively. We use tags $\{$SUBSET-$i \mid 1 \leq i \leq 3\}$ to denote them. The four subsets are used as training sets train 4 Parallel WaveGAN models $\{$SS$i \mid 0 \leq i \leq 3\}$. The training processes of the four models do however share the same validation set, since validation data has no bearing on model parameters. We then train the extraction–classification network described in section 3.3 to distinguish between audio waveforms from 5 classes: $\{$REAL, SS$i \mid 0 \leq i \leq 3\}$. To this end, we sample 50,000 waveform files from each source for training and 5,000 files for validation.

Table 3 and Figure 5 (see appendix) display the classification results and confusion matrix, respectively. The results obtained from the experiment corroborate our hypothesis that fingerprints are vocoder-specific on a finer granularity than architecture difference. Furthermore, it is shown that even a minor difference in vocoder model training (as is the case in SS0 vs. SS1, where the training sets differ by only one sample) can result models that produce sufficiently different fingerprints as to allow for distinguishing between audio generated by the two models.

Interestingly, if one were to use classification accuracy as the sole metric to judge the separability between the fingerprints, then it would seem that the magnitude of difference in training data has no meaningful impact on separability. Two models with training sets differing by 1 sample (accounting for 0.002% of the training set) still produced easily distinguishable fingerprints.

## 4.5 IMPACT OF INITIALIZATION SEEDS

In our third experiment, we investigate an even smaller variance factor: that of the initialization seeds. To accomplish this, we limit the models to the Parallel WaveGAN architecture, just as in the second experiment discussed above; we also use the same training data to train all models, and instead manually vary the seeds used in the initialization of model parameters. Specifically, we use seeds $s \in \{1000, 1001, 1002, 1003\}$ to initialize 4 models that we tag as $\{$SEED-$s\}$, which are then trained with the entirety of the LibriTTS clean speech subset. We then train the classifier to distinguish between 5 classes: $\{$REAL, SEED-$s \mid s \in \{1000, 1001, 1002, 1003\}\}$. We again sample 50,000 waveform files from each source for training and 5,000 files for validation.

Table 6 and Figure 6 (see appendix) display the classification results and confusion matrix, respectively. These results show that, even when models share the same architecture and are trained with the same data, the variation of initialization seeds can still result in sufficiently different fingerprints

for distinguishing between audio generated by the models. Given that the weight parameters of the vocoder models are supposed to converge towards a configuration that minimizes the objective function, our observation from this experiment entails that differences in vocoder weights can result in different fingerprints being produced by the models.

## 4.6 IMPACT OF TRAINING HYPERPARAMETERS

Our fourth experiment reported in this paper seeks to explore the impact of training hyperparameters that govern the training procedure, while fixing the architecture, training data, and initialization seeds of the vocoder models. We again limit the models to the Parallel WaveGAN architecture, with initialization seed as 1000 for all models, and use the entire clean training set of LibriTTS to train the vocoder models.

For the purposes of this experiment, we are interested specifically in the impact of adjusting the dropout rate and the batch size in training. We copy the PARALLEL-WAVEGAN-SEED-1000 model from the previous experiment and retag it as the baseline. This baseline does not use dropout, and each batch contains 6 samples. We then train four more models with the following parameters:

The last two models are trained for 150% and 75% of the number of steps as the baseline model, respectively, to compensate for the batch size influencing the total amount of data. Any unmentioned hyperparameter is kept the same as in the baseline model.

We then train the classifier to distinguish between 6 classes: {REAL, BASE, 1DROPOUT, 2DROPOUT, 4BATCH, 8BATCH}. As with all other experiments, we again sample 50,000 waveform files from each source for training and 5,000 files for validation.

Table 7 and Figure 7 (see appendix) display the classification results and confusion matrix, respectively. These results show that, even when models share the same architecture and are trained with the same data and initialization seeds, the precise manner of training can result in different models producing fingerprints that are different enough for distinguishing between audio generated by the models. It would therefore seem that the vocoder fingerprint is volatile to training.

## 4.7 COMPARISON OF ARCHITECTURAL AND PARAMETER DIFFERENCES

We conduct our last experiment to investigate the hypothesis that, even though differences in vocoder model parameters are sufficient to produce distinguishable fingerprints, differences in architectures predominate differences in parameters. In other words, if we have a fingerprint extraction–classification model that can detect fingerprints produced by vocoders of different architectures (such as the one presented in Experiment 1), it should still be able to reliably classify between these same architectures even when the vocoders have different parameters.

We train vocoders of 4 architectures: Parallel WaveGAN, HiFiGAN, Style MelGAN and Multiband MelGAN, denoted as P, H, S, and M, respectively. We train 3 vocoders for each architecture: The first with an initialization seed of 1000, the second with an initialization seed of 1001 and the third trained with a batch size of 8. The vocoders are trained with the LibriTTS dataset. The training setups of the models are summarized in Table 9.

We apply copy synthesis on the WSJ0 dataset (Garofolo et al., 2007). As the dataset comes in .sph files, these files are transformed into .wav through sph2pipe[1], resampled to 24 kHz, and randomly split into training–validation–evaluation sets with a 64%/16%/20% ratio. The training set is fed into models P0, H0, S0, and M0, and the results are labelled based on the vocoder architecture and used as the training set to train the Res2Net classifier. The validation set is fed into all models listed in Table 9, labelled with the architecture of the vocoder, and used as the validation set for the Res2Net classifier. The evaluation set is fed into the vocoders that were not involved in the training set. The copy-synthesis setup is summarized in Table 8.

Table 10 and Figure 8 (see appendix) display the classification results and confusion matrix. The results obtained from the experiment corroborate our hypothesis that, while fingerprints are vocoder-specific on a finer granularity than architecture difference (see Experiments 2–4), these differences are also less pronounced than architectural differences. As our experiments reported here deals

---

[1] https://github.com/robd003/sph2pipe

Table 8: Vocoder models used for copy synthesis in our final experiment.

| Subset | Parallel WaveGAN | HiFiGAN | Style MelGAN | Multiband MelGAN |
|---|---|---|---|---|
| training | P0 | H0 | S0 | M0 |
| validation | P0, P1, P2 | H0, H1, H2 | S0, S1, S2 | M0, M1, M2 |
| evaluation | P1, P2 | H1, H2 | S1, S2 | M1, M2 |

Table 9: Vocoder model setups in our final experiment.

| Model | Architecture | Seed | Batch | Steps |
|---|---|---|---|---|
| P0 | Parallel WaveGAN | 1000 | 6 | 500,000 |
| P1 | Parallel WaveGAN | 1001 | 6 | 500,000 |
| P2 | Parallel WaveGAN | 1000 | 8 | 375,000 |
| H0 | HiFiGAN | 1000 | 16 | 1,000,000 |
| H1 | HiFiGAN | 1001 | 16 | 1,000,000 |
| H2 | HiFiGAN | 1000 | 8 | 2,000,000 |
| S0 | Style MelGAN | 1000 | 16 | 1,000,000 |
| S1 | Style MelGAN | 1001 | 16 | 1,000,000 |
| S2 | Style MelGAN | 1000 | 8 | 2,000,000 |
| M0 | Multiband MelGAN | 1000 | 16 | 500,000 |
| M1 | Multiband MelGAN | 1001 | 16 | 500,000 |
| M2 | Multiband MelGAN | 1000 | 8 | 1,000,000 |

Table 10: Results (%) on the combined effects of vocoder architectures and training setups

| | Precision | Recall | F1 score |
|---|---|---|---|
| Real | 99.15 | 100.00 | 99.57 |
| P | 83.80 | 96.16 | 89.56 |
| H | 89.53 | 90.74 | 90.13 |
| M | 90.25 | 81.64 | 85.73 |
| S | 99.27 | 92.35 | 95.68 |
| Average | 91.65 | 91.31 | 91.31 |

solely with the acoustic features that depend on the specific vocoders being used to generate a given audio sample, we believe that these results offer much insight; namely that any fine-grained difference resulting from different training setups may be overshadowed by the differences in vocoder architecture, in the process of vocoder fingerprint analysis. The precise relationship between architecture and parameters, however, remains an objective to be investigated in our future efforts.

## 5 VISUALIZATION

While we observe the effectiveness of fingerprints in distinguishing between audio generated by different vocoder models through evaluating the classification performances, it is also possible to directly see the separability of these fingerprints through the t-SNE method (van der Maaten & Hinton, 2008) that maps from a space with higher dimensionality (in our case, 256) to a 2D surface while making similar points closer and dissimilar points farther on the new surface. Figure 3 (see Appendix) shows the results of t-SNE mapping of the fingerprints extracted in our four experiments. While minor overlaps exist, corresponding to the confusions in classification, the mapped representations still clearly show that fingerprints belonging to different vocoder models are distinguishable. Furthermore, the fact that the t-SNE algorithm naturally clusters fingerprints of the same model together indirectly confirms that vocoder fingerprints are model-specific, in that fingerprints of each model forms a concentrated distribution around different mean values.

## 6 CONCLUSION

We have presented our study of extracting vocoder fingerprints as an effort to identify fake audio generation source. Our five experiments verify the existence of model-specific vocoder fingerprints and demonstrate that even small differences in the training process of vocoder models can result in their fingerprints being distinct, allowing for fine-grained source identification of fake audio. Furthermore, we show that it is possible to determine the architecture of a vocoder used to generate an audio sample, even if the precise parameters of the said vocoder model has been changed because of a different training setup. We find this result encouraging because it suggests that our proposed usage of vocoder fingerprint extraction for forensics and copyright protection may be viable. Despite our results, we have yet to study the impact of different languages (and therefore different phonological feature distributions) and acoustic models can have on the fingerprints of generated audios, and exploring these questions will form the focus of our future studies.

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

## APPENDIX

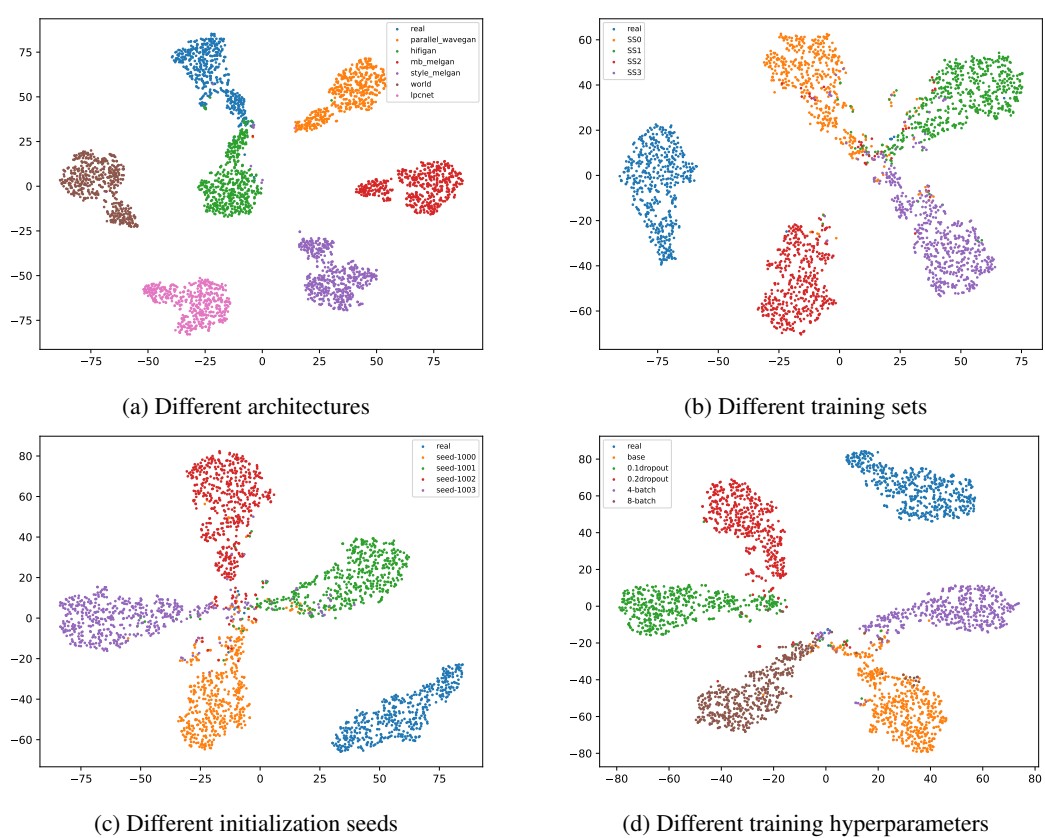

(a) Different architectures

(b) Different training sets

(c) Different initialization seeds

(d) Different training hyperparameters

Figure 3: t-SNE representation of fingerprints extracted in our experiments.

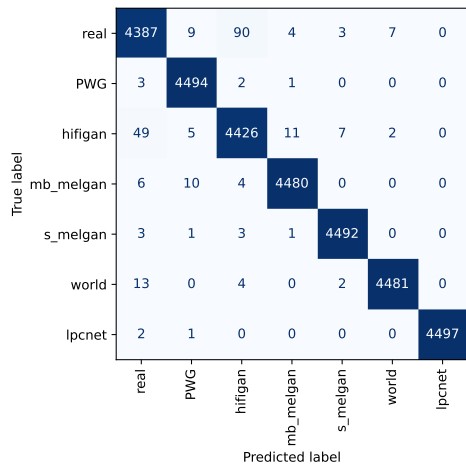

Figure 4: Confusion matrix on the impact of vocoder architectures

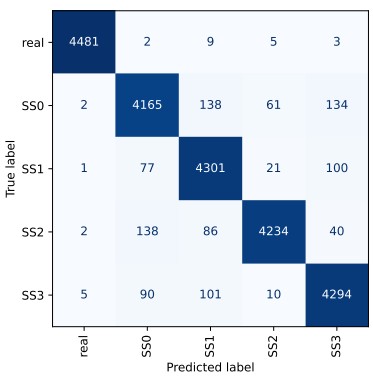

Figure 5: Confusion matrix on the impact of training data

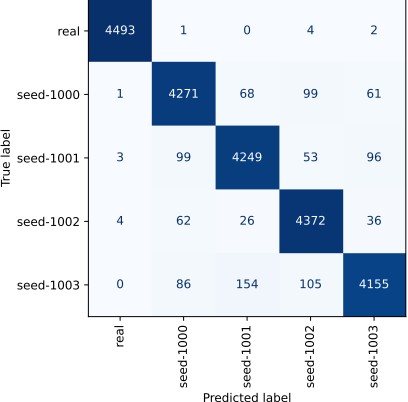

Figure 6: Confusion matrix of Experiment 3 (varied initialization seeds)

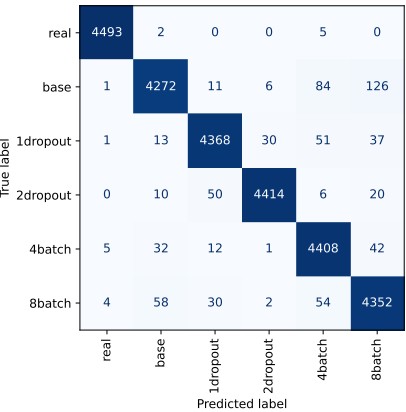

Figure 7: Confusion matrix of Experiment 4 (varied training hyperparameters)

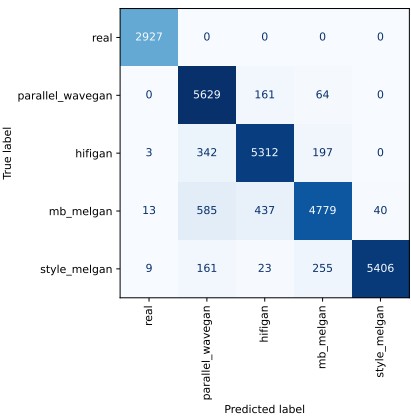

Figure 8: Confusion matrix of Experiment 5 (varied architecture and training hyperparameters)

