# OpenReview forum: "How Distinguishable Are Vocoder Models? Analyzing Vocoder Fingerprints for Fake Audio"
_ICLR.cc/2023/Conference — Submitted to ICLR 2023_

### Official Review · Reviewer_k7xs · 2022-10-31

**Confidence:** 3
**Correctness:** 2
**Technical Novelty And Significance:** 2
**Empirical Novelty And Significance:** 3
**Recommendation:** 3

**Clarity, Quality, Novelty And Reproducibility:**

**Clarity:** the paper is clearly written.

**Novelty:** the idea and analysis are novel.

**Reproducibility:** the authors provide all the details to reproduce the results.

**Strength And Weaknesses:**

**Strengths:**
1) this is an important topic and the analysis is interesting.
2) the paper is clearly written and easy to follow.

**Weaknesses:**
although the analysis is interesting:
1) some important and basic baselines are missing.
2) some of the claims in the paper should be adjusted / better motivated (see below).

**Summary Of The Paper:**

This paper analyzes the ability to identify the source model that generated the audio files. The authors explore generating samples from seven models different models. Specifically, the authors trained a classification network to distinguish between the samples. Results suggest that such classifiers can almost perfectly distinguish between the different vocoders.
The authors additionally provide an analysis of different training configurations such as batch_size, seed, dropout rate, and dataset split. Authors found that all such changes result in samples that are distinguishable from each other via a classification network.

**Summary Of The Review:**

As recently a large number of neural vocoders were proposed, the idea of analyzing them, specifically from the point of view of vocoder fingerprints is important and interesting research.
However, I still have some issues/questions:
1) did the authors analyze the same recording recorded from two different microphones? for instance, using WSJ0? as this research analyses the acoustic features of the generated audio, I would expect to see the same trend also for different mics.
2) the authors said that such findings allow source identification of fake audio, however, I'm not sure how. The authors show that even the smallest change in the training pipeline causes a completely different set of samples, so how can we know if given samples were generated from model-1 using different seeds or from model-2? Generally speaking, although the findings are interesting, it is not clear how to interpret that. Can we remove such behavior? Do we want/need to fix it? Can we identify if a sample was generated from model-1 also when considering different variations of the model? I believe all these questions should be discussed here in the paper.
3) Res2Net model was proposed for images, did you re-train the classifier on audio? If so, the visualizations in Figure 1 and Figure 7 are not very surprising, or am I missing something? These features are used for a linear classifier, which according to the results works pretty well. So it makes sense that these features are linearly separable.

I'm willing to increase my score in case I missed something or if the authors will address the concerns raised above.

---

> ### Author Response · Authors · 2022-11-18
> **Response to Review**
>
> > Q1: Did the authors analyze the same recording recorded from two different microphones? for instance, using WSJ0? as this research analyses the acoustic features of the generated audio, I would expect to see the same trend also for different mics.
>
> A1: We did not, at first, consider the potential acoustic feature differences introduced by different microphones in real audio. To address this oversight, in our newest experiment (section 4.7) aiming to investigate how differences in architecture and training setup interact, we opt to use the WSJ0 dataset for copy synthesis to train the classifier. It can be seen that the differences between microphones are not significant enough, when compared to the acoustic features of audio generated by different vocoders, to impede classification of vocoder architecture. While we have not yet had the opportunity to retroactively apply this to our already-presented experiments, we will take into consideration the potential influence from recording apparatuses in our future studies.
>
> > Q2: The authors said that such findings allow source identification of fake audio, however, I'm not sure how. The authors show that even the smallest change in the training pipeline causes a completely different set of samples, so how can we know if given samples were generated from model-1 using different seeds or from model-2? Generally speaking, although the findings are interesting, it is not clear how to interpret that. Can we remove such behavior? Do we want/need to fix it? Can we identify if a sample was generated from model-1 also when considering different variations of the model? I believe all these questions should be discussed here in the paper.
>
> A2: It was indeed an oversight on our part that our existing experiments, as they stand, fail to consider the relationship between architectural differences and parameter differences. To address this oversight, we conducted a new experiment (see section 4.7) that show the relationship between differences in architecture and differences in parameters. Judging by the results obtained from the experiment, we therefore amend our conclusion by pointing out that, while different training setups do indeed lead to differences in fingerprints, such differences are not significant to the same magnitude as architectural differences. Although, as the results show, unknown changes to parameters of a known-architecture vocoder will in fact disturb the fingerprints enough to make the classification less accurate. Following the addition of a final experiment, we have revised our manuscript accordingly.
>
> > Q3: Res2Net model was proposed for images, did you re-train the classifier on audio? If so, the visualizations in Figure 1 and Figure 7 are not very surprising, or am I missing something? These features are used for a linear classifier, which according to the results works pretty well. So it makes sense that these features are linearly separable.
>
> A3: The Res2Net model in our experiment was indeed retrained on audio (more precisely, on LFCC features extracted from audio waveforms). In essence, the motivation behind choosing Res2Net was to obtain multi-scaled features of different resolutions, and investigate if those features can be linearly separable, and our results show that they are. Figures 1 and 7 are meant to demonstrate in a more intuitive manner the separability of those features.

---

### Official Review · Reviewer_S3gk · 2022-11-02

**Confidence:** 4
**Correctness:** 3
**Technical Novelty And Significance:** 1
**Empirical Novelty And Significance:** 2
**Recommendation:** 3

**Clarity, Quality, Novelty And Reproducibility:**

**Clarity and Quality:**

This paper is organized well and presented clearly.

**Novelty:**

As mentioned above, the novelty of this work is limited.

**Reproducibility:**

Some details for implementing and training the vocoders are missing. Note that if some vocoder is not well converged and well optimized, the classifier can easily identify it according to specific distortion in generated utterances. Below are some examples of missing details.

- What were the acoustic features used for the vocoders? What were the parameters to extract the features?

- The original LPCNet restores waveforms based on self-defined acoustic features (BFCC and others). It is said that the LPCNet in this work used Mel spectrums. Why?

- How were the vocoders trained? Did the training configurations (e.g., steps, batch size) the same as their original papers? Or were they set the same for training different vocoders?

In Section 4.2, for the fifth architecture, does it mean "Multi-band WaveGAN" or "Multi-band MelGAN"?


**Strength And Weaknesses:**

**Strength:**

This work analyzed the distinguishability of different vocoders from four aspects and showed the existence of distinct vocoder fingerprints in these four scenarios.

**Weaknesses:**

The novelty and significance of this work are limited.

Discovering fingerprints of vocoders with different architectures has been studied by Yan et al., and the novelty of this work is extending the analysis to other aspects, such as training data, initial weights, and batch sizes.

Although the experimental results show the existence of distinct vocoder fingerprints under different scenarios, the lack of further analysis and discussion makes the results less significant and explainable. To better support the results, the author should provide more insight, explain factors that cause distinct fingerprints, or show how the classifier distinguishes different vocoders.

Xinrui Yan, Jiangyan Yi, Jianhua Tao, Chenglong Wang, Haoxin Ma, Tao Wang, Shiming Wang, and Ruibo Fu. An Initial Investigation for Detecting Vocoder Fingerprints of Fake Audio. arXiv:2208.09646 [cs, eess], August 2022. doi: 10.1145/3552466.3556525.


**Summary Of The Paper:**

This paper aims to verify the existence of vocoder fingerprints by training a classifier to identify the sources of generated audio waveforms. The authors analyzed the distinguishability of different vocoders from four aspects: (1) vocoders with different architectures, (2) vocoders trained on different datasets, (3) vocoders with different initial weights, and (4) vocoders with different batch sizes and dropout rates. The experimental results show that with different architectures or even small differences in the training process, the fingerprints of different vocoders are distinct. All different vocoders are distinguishable under the four scenarios mentioned above.

**Summary Of The Review:**

Overall, given the lack of novelty, further analysis on factors that cause distinct fingerprints, and experimental details, the contribution, significance, and reproducibility of this work are limited.

---

> ### Author Response · Authors · 2022-11-18
> **Response to Review**
>
> > Q1: What were the acoustic features used for the vocoders? What were the parameters to extract the features?
>
> A1: We follow the setup instructions in ParallelWaveGAN and attempt to unify the spectrum input to the different GAN vocoder models, as to remove the influence of the input features in the vocoders. The GAN models all take 80-band mel-spectra as inputs, with a frame length of 50 ms and a hop length of 12.5 ms; as a result, these features are pre-extracted and remain the same for all GAN models.
> The LPCNet vocoder (with its official implementation at https://github.com/xiph/LPCNet) is trained directly on the raw waveforms, since it has built-in feature extractors.
>
> > Q2: How were the vocoders trained? Did the training configurations (e.g., steps, batch size) the same as their original papers? Or were they set the same for training different vocoders?
>
> A2: We generally follow the training configurations presented in the original papers (except in Experiment 4, where the training configurations like batch sizes and steps are the variables of study; in those cases, the deviation from the original paper for ParallelWaveGAN have been mentioned).
>
> Specifically, the LPCNet vocoder in Experiment 1 is trained with batch size of 128 for 100 epochs, with the AMSGrad optimization method presented in its original paper; the ParallelWaveGAN vocoders are trained with batch size of 6 for 500,000 steps with the Adam optimizer; the HiFiGAN and Multiband MelGAN vocoders are trained with batch size of 16 for 1 million steps with the Adam optimizer; Style MelGAN with batch size of 16 for 500,000 steps with the Adam optimizer.
>
> We decide not to force every vocoder to be trained with precisely the same configurations because each “step” in the training process contributes to the convergence of model parameters differently, depending on the precise architecture used, and as such it does not make much sense to us to force uniformity in numbers, and instead we defer to the empirical judgment of the respective authors proposing. Naturally, in Experiments 2-4, where the vocoders have the same architecture (Parallel WaveGAN), the training configurations are the same across the board (except where otherwise indicated, as is the case for Experiment 4).

---

### Official Review · Reviewer_4uuW · 2022-11-03

**Confidence:** 4
**Correctness:** 3
**Technical Novelty And Significance:** 2
**Empirical Novelty And Significance:** 2
**Recommendation:** 5

**Clarity, Quality, Novelty And Reproducibility:**

This paper lacks related works and an explanation of why those works are related to this paper. Also, no related works are mentioned to support the observation from the experiment results.

The novelty is limited. Compared to the previous work (Yan et al. 2022), this paper further extends the experiments to study the effect of hyperparameters and training data. However, the methodology is similar to the previous work, and not much deeper insight is provided in this paper.

No source code and detailed experiment setup are provided. Therefore, the experiment result might not be easy to reproduce.


**Strength And Weaknesses:**

### Strength:
The research topic of this paper is recently emerging and interesting.

Unlike standard fake speech detection, which is mainly a binary classification problem, this paper further studies whether we can tell which vocoder is used for fake speech synthesis.

### Weaknesses:
The authors mainly showed the experiment result without further investigation. For example, why the vocoder training is so sensitive to the training data, hyperparameter, and random seed? (Even differences in one data point or the random seed lead to significant differences)

There’s not much insight into the usage scenario of the proposed fingerprint extraction and classification pipeline. How can the fingerprint of the vocoders be used in a real-world scenario? How to use the fingerprint of the vocoders to protect the copyright? And how can the fingerprint of the vocoders provide benefits to fake speech detection?

Only “Copy synthesis” is studied in this paper. The paper mentioned vocoders are used in speech synthesis technology, such as TTS and VC, and therefore the fingerprint of the vocoder is important to fake speech detection. However, whether the fingerprint extraction is indeed affected by the TTS or VC pipeline is not studied.

This paper only studies the classification in the in-domain scenarios. That is, the vocoders are given, and the training data is limited to LibriTTS. However, in the real-world scenario, one cannot know and list all the possible vocoders. As a result, a simple classification experiment might not be enough. The authors can consider including the “unknown” category or computing the similarity between each vocoder.


**Summary Of The Paper:**

This paper studies the fingerprints of the vocoders. The authors studied whether resynthesized speech can be classified into real speech and a set of vocoders. Specifically, A deep classifier is trained on top of the LFCC feature extracted from the synthesized waveform. In this paper, 7 vocoders are studied in the experiments: WORLD, LPCNet, Parallel WaveGAN, HiFiGAN, Multi-band WaveGAN, and Style-MelGAN.
The experiment result shows that different vocoders have their own fingerprint, and the fingerprint is sensitive to training hyperparameters and the vocoder architecture.


**Summary Of The Review:**

Overall speaking, this paper mainly provided the experiment results without further investigation and explanation, which might not be inspiring enough to motivate the community. And therefore, the paper is not ready to publish without deeper investigation.

---

> ### Author Response · Authors · 2022-11-18
> **Response to Review**
>
> > Q1: The authors mainly showed the experiment result without further investigation. For example, why the vocoder training is so sensitive to the training data, hyperparameter, and random seed? (Even differences in one data point or the random seed lead to significant differences)
>
> A1: In our experiments seeking to investigate these differences, we trained Res2Net classifiers specifically to extract these details from the generated audio. Indeed, the differences in waveforms are significant enough to still result in a separable difference in their LFCC representations, but our pre-revision experiments, as they stand, do not really address the combined effects of architecture and parameter differences.
>
> To this end, we conducted a new experiment (see section 4.7) that show the relationship between differences in architecture and differences in parameters. Judging by the results obtained from the experiment, we therefore amend our conclusion by pointing out that, while different training setups do indeed lead to differences in fingerprints, such differences are not significant to the same magnitude as architectural differences.
>
> > Q2: There’s not much insight into the usage scenario of the proposed fingerprint extraction and classification pipeline. How can the fingerprint of the vocoders be used in a real-world scenario? How to use the fingerprint of the vocoders to protect the copyright? And how can the fingerprint of the vocoders provide benefits to fake speech detection?
>
> A2: We recognize the seemingly inadequate exploration of potential usage scenario of the extraction-classification pipeline, partially due to the fact that we mostly focused on the role of vocoders in producing these fingerprints. It is true that TTS and VC procedures involve more than the vocoder, and such, vocoder fingerprint extraction in and of itself may not be sufficient. However, in the cases where the vocoder and the acoustic models of a pipeline are jointly trained and optimized, the parameters of that vocoder may differ from a pre-trained vocoder model of the same architecture. We believe that the parameter differences in vocoders, as demonstrated in our experiments, can still be leveraged to protect copyright in these cases.
>
> As for the benefits to fake speech detection, we believe it comes mostly in two areas:
> 1. It offers some interpretability to the process of fake speech detection. If, in addition to the binary classification of real/fake, we can furthermore pinpoint the origin of the fake audio, it may lend more credibility to the detection results.
> 2. On the flipside, it may also expose an aspect of TTS and VC that could be improved, so as to generate audio that is less detectable through fingerprint extraction such as the procedure we present here.
>
> > Q3: Only “Copy synthesis” is studied in this paper. The paper mentioned vocoders are used in speech synthesis technology, such as TTS and VC, and therefore the fingerprint of the vocoder is important to fake speech detection. However, whether the fingerprint extraction is indeed affected by the TTS or VC pipeline is not studied.
>
> A3: This is indeed true. Our motivation behind using copy synthesis was to fix the feature inputs to the different vocoders, to better focus on the fingerprints of these vocoder models in the initial stages of our investigation. If we were to introduce both vocoder models and TTS/VC pipelines as variables, it would be hard to pinpoint how much fingerprint is left by the vocoder. While we recognize this issue, with the finite time and effort so far, we have been unable to address all the issues we found, and therefore we focus on vocoders in our experiments presented here. In recognition of the insufficiency of scope, we plan to address the impact of TTS and VC pipelines in our future work.
>
> > Q4: This paper only studies the classification in the in-domain scenarios. That is, the vocoders are given, and the training data is limited to LibriTTS. However, in the real-world scenario, one cannot know and list all the possible vocoders. As a result, a simple classification experiment might not be enough. The authors can consider including the “unknown” category or computing the similarity between each vocoder.
>
> A4: This is a very astute suggestion, one that we will for sure integrate into our future work in this direction. In the meantime, our newest experiment appears to corroborate the intuition that different vocoders of the same architecture exhibit more similarities than vocoders of different architectures. In the future we will seek to measure such similarity so that it can be better used for fingerprint extraction purposes.

---

### Author Response · Authors · 2022-11-18
**Response to All Reviews**

Dear reviewers:

We would like to, first and foremost, present our deepest gratitude for your time and in reading our manuscript, as well as for the comments and critiques. We have modified our manuscript accordingly (significant revisions have been highlighted). Please find our responses to your questions, concerns, and critiques below.

To address the reproducibility issue, we have released the codebase we used as the fingerprint extraction network, zipped in the supplementary materials file.

---

### Decision · Program_Chairs · 2023-01-20

**Decision:**

Reject

**Justification For Why Not Higher Score:**

The paper lacks novelty and new insight, so there is no good reason to accept.

**Justification For Why Not Lower Score:**

N/A

**Metareview: Summary, Strengths And Weaknesses:**

This paper aims to verify the existence of vocoder fingerprints by training a classifier to identify the sources of generated audio waveforms. The authors analyzed the distinguishability of different vocoders from architectures, datasets, initial weights, and batch sizes and dropout rates. The experimental results show that the fingerprints of different vocoders are distinct. All different vocoders are distinguishable under the scenarios mentioned above. But after the rebuttal, the authors further add an experiment to show that the vocoder architecture differences may overshadow fine-grained differences resulting from different training setups in the vocoder fingerprint analysis process.

Discovering fingerprints of vocoders with different architectures has been studied by previous work, and the novelty of this work is extending the analysis to the effect of hyperparameters and training data. However, the methodology is similar to the previous work, and not much deeper insight is provided in this paper.